# Exercise Alleviates Cognitive Functions by Enhancing Hippocampal Insulin Signaling and Neuroplasticity in High-Fat Diet-Induced Obesity

**DOI:** 10.3390/nu11071603

**Published:** 2019-07-15

**Authors:** Hye-Sang Park, Sang-Seo Park, Chang-Ju Kim, Mal-Soon Shin, Tae-Woon Kim

**Affiliations:** 1Department of Kinesiology, College of Public Health and Cardiovascular Research Center, Lewis Katz school of Medicine, Temple University, Philadelphia, PA 19122, USA; 2Department of Physiology, College of Medicine, KyungHee University, Seoul 02447, Korea; 3School of Global sport studies, Korea University, Sejong 30019, Korea; 4Exercise Rehabilitation Research Institute, Department of Exercise & Health Science, Sangmyung University, Seoul 03016, Korea

**Keywords:** exercise, obesity, cognitive function, hippocampus, insulin signaling, neuroplasticity

## Abstract

Obesity, caused by a high-fat diet (HFD), leads to insulin resistance, which is a precursor of diabetes and a risk factor for impaired cognitive function, dementia, and brain diseases, such as Alzheimer’s disease. Physical exercise has positive effects on obesity and brain functions. We investigated whether the decline in cognitive function caused by a HFD could be improved through exercise by examining insulin signaling pathways and neuroplasticity in the hippocampus. Four-week-old C57BL/6 male mice were fed a HFD or a regular diet for 20 weeks, followed by 12 weeks of treadmill exercise. To ascertain the effects of treadmill exercise on impaired cognitive function caused by obesity, the present study implemented behavioral testing (Morris water maze, step-down). Moreover, insulin-signaling and neuroplasticity were measured in the hippocampus and dentate gyrus. Our results demonstrated that HFD-fed obesity-induced insulin resistance was improved by exercise. In addition, the HFD group showed a decrease in insulin signaling and neuroplasticity in the hippocampus and the dentate gyrus and increased cognitive function impairment, which were reversed by physical exercise. Overall, our findings indicate that physical exercise may act as a non-pharmacologic method that protects against cognitive dysfunction caused by obesity by improving hippocampal insulin signaling and neuroplasticity.

## 1. Introduction

Obesity is closely associated with the risk of cardiovascular complications, diabetes, and hypertension, and it damages the brain by affecting the chemical and risk factors associated with pathological neurodegeneration [1]. Therefore, obesity is a risk factor for brain disorders, including declining cognitive function, dementia, and Alzheimer’s disease [2,3]. In fact, obesity in old age is associated with structural changes in brain volume [4], and obesity in middle age independently increases the risk of dementia [5].

In particular, obesity induces insulin resistance, which is a preliminary stage of diabetes, through various mechanisms, including insulin overproduction due to excessive nutritional intake, decreased insulin receptor (IR) function, and/or deficit in the intracellular signaling system. The primary function of insulin in peripheral tissues is glucose homeostasis; however, it has somewhat more diverse actions in the central nervous system (CNS). In the CNS, the direct role of insulin/IR signaling is to improve cognitive functions, including learning and memory [6]. Thus, the brain is an important target for insulin and IR. Damage to insulin signaling in the brain has been suggested to be related to neurodegenerative disease [7], and some patients with type-II diabetes have exhibited impaired cognitive functions [8]. Insulin has been proven essential to optimal hippocampal memory processes, and insulin resistance in the telencephalon can result in cognitive deficiencies, which usually accompany type-II diabetes [9]. 

A high-fat diet harms the structure and function of the hippocampus, which plays an important role in learning and memory, and decreases the levels of brain-derived neurotrophic factor (BDNF) and neurogenesis in the hippocampus [10,11]. Neurogenesis of the adult brain occurs in the subgranular zone and neurons migrate into the granule cell layer of the hippocampal dentate gyrus and become dentate granule cells [12]. BDNF controls neural development, survival, and differentiation through tyrosine protein kinase B (trkB) receptors, and, unlike other neurotrophic factors, BDNF secretion is increased by activity-dependent modulators [13]. Hippocampal BDNF is not only related to cognitive functions, including learning and memory, but is also associated with insulin resistance, and it also improves glucose homeostasis and insulin resistance in obese, diabetic mice by decreasing food intake and blood glucose levels [14]. Especially, dysfunctions in the hippocampal insulin signaling might be responsible for cognitive impairment [15]. The significant risk factors for the preliminary diabetic stage of insulin resistance and type-II diabetes, such as obesity and physical inactivity are associated with alterations in the brain and increase the risk of developing dementia [16,17,18]. Thus, obesity results in damage to insulin signaling pathways, deficiencies in neuroplasticity, and insulin resistance, which then leads to type-II diabetes in the periphery and Alzheimer’s disease, which has been called type-III diabetes, in the CNS. Here, we used a diet-induced model rather than a genetic model to model obesity. The diet-induced obesity model is similar to humans with obesity and hyperglycemia, total cholesterol, and insulin resistance [19].

Exercise is a strongly recommended therapy and preventative measure for patients with obesity and those with insulin resistance caused by obesity.

Accordingly, in our study, we induced cognitive dysfunction through obesity that resulted from a high-fat diet (HFD) and investigated whether the decline in cognitive function could be improved with exercise by examining insulin signaling pathways and neuroplasticity in the hippocampus.

## 2. Materials and Methods

### 2.1. Animals and Diet

All animal experimental procedures conformed to the regulations stipulated by the National Institutes of Health (NIH) and the guidelines of the Korean Academy of Medical Science. This study was approved by the Kyung Hee University Institutional Animal Care and Use Committee (Seoul, Korea) (KHUASP (SE)-14-018). Male 4-week-old C57BL/6 mice were randomly divided into four groups (n = 20 per group): control (CON), CON and exercise (EX), high-fat diet (HFD), and HFD and EX. The HFD chow contained 60% fat and was provided ad libitum.

### 2.2. Exercise Protocol

Treadmill exercise started 20 weeks after the intake of the HFD. The exercise group began exercising on a treadmill made for animal use. During the exercise, animals performed 5 min of warm-up at a 0° inclination at 3 m/min, 30 min of the main exercise at 10 m/min, and 5 min of cool down at 3 m/min for the first 2 weeks. Following this period, animals performed 40 min of the main exercise at 10 m/min during weeks 3~4, 30 min of the main exercise at 13 m/min during weeks 5~6, 40 min of the main exercise at 13 m/min during weeks 7~8, 40 min of the main exercise at 16 m/min during weeks 9~10, and 50 min of the main exercise at 16 m/min during weeks 10~12. The exercise was performed once a day and six days per week for 12 consecutive weeks. Electrical stimulation was removed during treadmill running to minimize stress.

### 2.3. Glucose Tolerance Test

We administered an intraperitoneal injection of glucose (1.5 g/kg) to 18 h fasted mice after 20 weeks of a HFD and 12 weeks of treadmill exercise. Blood glucose levels were measured before and at 15, 30, 45, 60, 120, and 180 min after glucose injections. Blood was collected from the animals’ tails.

### 2.4. Step-Down Avoidance Task

The latency time of the step-down avoidance task was determined to evaluate long-term memory. 24 h after training, the latency time (s) in each group was measured. The mice were placed on a 7 × 25 cm platform that was 2.5 cm in height. The platform faced a 42 × 25 cm grid of parallel stainless steel bars, 0.1 cm in caliber, spaced 1 cm apart. During the training sessions, the animals received a 0.5 mA scramble foot shock for 2 s immediately upon stepping down. The interval of time which elapsed between the rats stepping down and placing all four paws on the grid was defined as the latency time. A latency time >180 s was counted as 180 s. 

### 2.5. Morris Water Maze Task

The Morris water maze task was used to assess spatial learning and working memory. One day before training, the mice were habituated to swimming for 60 s in the pool without a platform. All mice were trained three times a day for five consecutive days. A probe trial was performed 24 h after the final training session. When a mouse found the platform, it was allowed to remain there for 30 s. If the mouse did not find the platform within 60 s, it was guided by hand to the platform. The mice underwent a 60 s retention probe test, after which the platform was removed from the pool. Data were automatically collected via the Smart Video Tracking System (Smart version 2.5, Panlab, Barcelona, Spain).

### 2.6. Immunofluorescence for 5-Bromo-2’-Deoxyuridine (BrdU)- and Neuronal Nucleic (NeuN)

Immunofluorescence was performed to detect BrdU/NeuN-positive cells in the dentate gyrus. In brief, the brain sections were permeabilized with 0.5% Triton X-100 in PBS for 20 min, incubated in 50% formamide-2× standard saline citrate at 65 °C for 2 h, denaturated in 2 N HCl at 37 °C for 30 min, and rinsed twice in 100 mM sodium borate (pH 8.5). The sections were incubated overnight with rat anti-BrdU antibody (1:500; Abcam, ab6326) and mouse anti-NeuN antibody (1:500; Millipore, MAB377). The brain sections were then washed in PBS and incubated with secondary antibodies, including anti-mouse IgG Alexa Fluor-488, anti-rabbit IgG Alexa Fluor-594, and anti-rat IgG Alexa Fluor-550, for 2 h. The sections with the coverslips were mounted using a fluorescent mounting medium (Dako Cytomation). Fluorescent images of the slides were captured using a confocal laser scanning microscope (LSM-700; Carl Zeiss, München-Hallbergmoos, Germany).

### 2.7. Immunohistochemistry for Doublecortin (DCX)

To visualize cell differentiation, we performed immunohistochemistry for DCX in the dentate of the hippocampus. The sections were incubated in PBS for 10 min, washed three times for 3 min in PBS, and incubated in 1% H_2_O_2_ for 15–30 min. The sections were selected from each brain and incubated overnight with goat anti-DCX antibody (1:500; Santa Cruz Biotech, sc-8066). The following day, the sections were incubated with biotinylated rabbit secondary antibody (1:250; Vector Laboratories) for 90 min. The secondary antibody was amplified with the Vector Elite ABC kit^®^ (1:100; Vector Laboratories). Antibody-biotin-avidin-peroxidase complexes were visualized using a DAB substrate kit (vector Laboratories). The slides were air-dried overnight at room temperature, and the coverslips were mounted using Permount ^®^.

### 2.8. Western Blot for BDNF, Tyrosine Protein Kinase B, and Insulin Signaling

Hippocampal tissue was homogenized on ice and lysed in a radioimmunoprecipitation (RIPA) cell lysis buffer (1X) with ethylenediaminetetraacetic acid (EDTA). Protein content was measured using a Bio-Rad colorimetric protein assay kit (Bio-Rad, Hercules). We separated 20 µg of protein on SDS-polyacrylamide gels and transferred the protein onto a nitrocellulose blotting membrane, which was incubated with the following primary antibodies: mouse β-actin (1:1000; Santa Cruz Biotech), rabbit BDNF (Santa Cruz Biotech, sc-20981), TrkB (1:1000; Santa Cruz Biotech, sc-119), IR (1:1000; cell signaling, #3095), p-IR (Tyr1162/1163) (1:1000; cell application, CB 486), PI3K (Santa Cruz Biotech, sc-1637), p-PI3K (Tyr467, sc-293115) (Santa Cruz Biotech, sc-293115) PDK (cell signaling, #3062), p-PDK (Ser241) (1:1000; cell signaling, #3061), AKT (Santa Cruz Biotech, sc-8312), p-AKT (Ser 473) (Santa Cruz Biotech, sc-135651), GSK-3β (cell signaling, #9315), and p-GSK3β (Ser 9) (1:1000; cell signaling, #9323). Horseradish peroxidase-conjugated anti-rabbit secondary antibody was used for BDNF, TrkB, IR, p-IR, PDK, p-PI3K, p-PDK, AKT, p-AKT, GSK-3β, and p-GSK-3β and anti-mouse secondary antibody was used for PI3K and β-actin. Experiments were performed under normal lab conditions and at room temperature (except for the transferred membrane).

### 2.9. Data Analysis

Following Western blotting, densitometry was performed to confirm the expression of BDNF, trkB, and insulin signaling using Molecular AnalystTM, version 1.4.1. The numbers of BrdU/NeuN- and DCX-positive cells in the dentate gyrus were counted hemilaterally under a light microscope (Olympus, Tokyo, Japan) and expressed as the number of cells per square millimeter in the dentate gyrus. The area of the dentate gyrus was measured using the Image-Pro^®^ Plus image analysis system (Media Cyberbetics Inc., Silver Spring, MD, USA). Data were analyzed using one-way ANOVAs, and Duncan post-hoc tests were conducted when applicable. All values were expressed as the mean ± standard error of the mean (S.E.M.), and a *p* value < 0.05 was considered significant.

## 3. Results

### 3.1. Effects of High-Fat Feeding and Exercise on Body Weight and Blood Glucose Level

After 20 weeks, the body weight of animals in the HFD group (50.02 ± 0.40 g) was significantly increased as compared with that of animals in the CON group who were on a regular diet (33.02 ± 0.38 g; *p* < 0.001). In the blood glucose, the HFD group significantly increased in fasting at 15, 30, 45, 60, 120, and 180 min after injections of glucose compared with those in CON group (*p* < 0.001, respectively). The body weight of animals in the HFD group was significantly increased compared with that of the animals in the CON group (*p* < 0.001), whereas the HFD and EX group were significantly decreased compared to that in the HFD group (*p* < 0.001). The body weight of animals in the CON and EX group (i.e., animals that performed 12 weeks of treadmill exercise after the intake of a regular diet), was significantly decreased as compared with that of animals in the CON group (*p* = 0.001). For blood glucose, the HFD group significantly increased in fasting and after injection of glucose 15, 30, 45, 60, 120, and 180 min compared to those in the CON group (*p* < 0.001, respectively), whereas the HFD and EX group significantly decreased in fasting and 15, 30, 45, 60, 120, and 180 min after injection of glucose compared to those in HFD group (*p* < 0.001, respectively) (Figure 1).

### 3.2. Effect of Treadmill Exercise on Memory

Results from the Morris water maze demonstrated that spatial learning was significantly decreased in the HFD group as compared with that in the CON group on day 3 of training (34.54 ± 1.88 s vs. 43.10 ± 1.82 s: *p* = 0.008); however, spatial learning was significantly increased in the HFD and EX group as compared with that in the HFD group (43.10 ± 1.82 s vs. 36.04 ± 2.04 s: *p* = 0.048). Similarly, on days 4 and 5 of training, there was a significant decrease in spatial learning in the HFD group as compared with that in the CON group (26.83 ± 1.69 s vs. 34.72 ± 1.63 s: *p* = 0.013; 16.52 ± 1.40 s vs. 25.00 ± 1.83 s: *p* = 0.001, respectively), while spatial learning was significantly increased in the HFD and EX group as compared with that in the HFD group (34.72 ± 1.63 s vs. 28.02 ± 2.00 s: *p =* 0.049; 25.00 ± 1.83 s vs. 19.75 ± 1.54 s: *p =* 0.035, respectively) (Figure 2a). Furthermore, spatial working memory was significantly decreased in the HFD group as compared with that in the CON group (29.61 ± 1.03% vs. 17.35 ± 0.94%: *p <* 0.001), and there was a significant increase in spatial working memory in the HFD and EX group as compared with that in the HFD group (17.35 ± 0.94% vs. 26.21 ± 0.86%: *p <* 0.001) (Figure 2b). Results from the step-down avoidance task demonstrated that there was a significant decrease in long-term memory in the HFD group as compared with that of the CON group (260.06 ± 14.74 s vs. 47. 47 ± 11.30 s: *p <* 0.001), and there was a significant increase in long-term memory in the HFD and EX group as compared with that in the HFD group (47.47 ± 11.30 s vs. 208.73 ± 24.63 s: *p <* 0.001) (Figure 2c).

### 3.3. Effect of Treadmill Exercise on Hippocampal Insulin Signaling

For analysis of our Western blotting data, we set each cascade of the CON group to 1.00, and our results demonstrated that the pIRß/t-IRß ratio, p-IRS-1/t-IRS-1 ratio, p-PI3K/t-PI3K ratio, p-PDK1/t-PDK ratio, p-AKT/t-AKT ratio, and p-GSK-3ß/t-GSK-3ß ratio were significantly decreased in the HFD group as compared with those in the CON group (1.00 ± 0.00 vs. 0.48 ± 0.02: *p <* 0.001; 1.00 ± 0.00 vs. 0.67 ± 0.06: *p =* 0.001; 1.00 ± 0.00 vs. 0.61 ± 0.05: *p <* 0.001; 1.00 ± 0.00 vs. 0.42 ± 0.03: *p <* 0.001; 1.00 ± 0.00 vs. 0.36 ± 0.05: *p <* 0.001; and 1.00 ± 0.00 vs. 0.69 ± 0.03: *p <* 0.001, respectively). Additionally, the pIRß/t-IRß ratio, p-IRS-1/t-IRS-1 ratio, p-PI3K/t-PI3K ratio, p-PDK1/t-PDK ratio, p-AKT/t-AKT ratio, and p-GSK-3ß/t-GSK-3ß ratio were significantly increased in the HFD and EX groups as compared with those in the HFD group (0.48 ± 0.02 vs. 0.64 ± 0.02: *p =* 0.044; 0.67 ± 0.06 vs. 0.90 ± 0.07: *p =* 0.042; 0.61 ± 0.05 vs. 0.81 ± 0.03: *p =* 0.001; 0.42 ± 0.03 vs. 0.57 ± 0.05: *p =* 0.009; 0.36 ± 0.05 vs. 0.60 ± 0.07: *p =* 0.001; and 0.69 ± 0.03 vs. 0.87 ± 0.03: *p =* 0.032, respectively) (Figure 3).

### 3.4. Effect of Treadmill Exercise on BDNF and TrkB Expression in the Hippocampus

When the levels of BDNF and trkB in the CON group were set to 1.00, the protein expression levels of BDNF and trkB were significantly decreased in the HFD group as compared with those in the CON group (1 ± 0.00 vs. 0.38 ± 0.04: *p <* 0.001; 1.00 ± 0.00 vs. 0.51 ± 0.04: *p <* 0.001, respectively), while the expression levels of BDNF and trkB were significantly increased in the HFD and EX group as compared with those in the HFD group (0.38 ± 0.04 vs. 0.63 ± 0.05: *p <* 0.001; 0.51 ± 004 vs. 0.73 ± 0.05: *p =* 0.003, respectively) (Figure 4).

### 3.5. Effect of Treadmill Exercise on Neurogenesis in the Hippocampal Dentate Gyrus

To examine neurogenesis in the hippocampus, we stained tissue with BrdU and NeuN. Results from immunohistochemistry demonstrated that the number of BrdU/NeuN-double-positive cells was significantly decreased in the HFD group as compared with that in the CON group (200.57 ± 9.56 cells/mm^2^ vs. 108.26 ± 12.27 cells/mm^2^: *p <* 0.001), while the number of BrdU/NeuN-double-positive cells was increased in the HFD and EX groups as compared with that in the HFD group (108.26 ± 12.27 cells/mm^2^ vs. 161.23 ± 16.25 cells/mm^2^: *p =* 0.027) (Figure 5). 

### 3.6. Effect of Treadmill Exercise on the Expression of DCX in the Hippocampal Dentate Gyrus

To examine cell differentiation in the dentate gyrus, we stained tissue with DCX. The number of DCX-positive cells was significantly reduced in the HFD group as compared with that in the CON group (403.34 ± 14.41 cells/mm^2^ vs. 287.07 ± 16.02 cells/mm^2^: *p <* 0.001), while the number of DCX-positive cells was significantly increased in the HFD and EX group as compared with that in the HFD group (287.07 ± 16.02 cells/mm^2^ vs. 354.48 ± 6.97 cells/mm^2^: *p =* 0.003) (Figure 6).

## 4. Discussion

Here, we investigated the effects of a HFD on cognitive function in male mice. Our results demonstrated that the body weight and blood glucose levels of animals that were fed a HFD were increased as compared with those of CON animals. Furthermore, findings from the step-down and Morris water maze behavioral paradigms revealed that the long-term memory, spatial working memory, and learning of animals in the HFD group were decreased as compared with the memory and learning of animals in the CON group. Research has suggested that the body mass index is independently associated with cognitive function [20], and even a short-term HFD may cause deterioration of hippocampus-dependent spatial learning and deficits in cognitive function before the onset of hyperlipidemia and hyperinsulinemia symptoms [21]. In the brain, insulin not only controls glucose and lipid metabolism but also affects neural development and activity, which play an important role in learning and memory [22]. Impairments in insulin regulation are associated with obesity, diabetes, cardiovascular disease, and hypertension [23], and abnormalities in neural insulin signaling pathways are associated with various neurodegenerative diseases and deficits in learning and memory [24]. IRs exist in various parts of the brain but are particularly expressed in high concentrations in the hypothalamus, hippocampus, and cerebral cortex [25]. Among these regions, the hippocampus is responsible for memory and other cognitive functions; therefore, it plays an important role in storing new memories. In recent studies, hippocampal insulin has been proposed to regulate cognitive functions [9], and obesity leads to impairments in hippocampus-dependent behavior that is caused by hippocampal insulin resistance. Previous studies have reported that IR signaling in the hippocampus is decreased in animal models of obesity [26]. This may be because hippocampal insulin/IR activity that is related to learning and memory directly regulates the activity of receptors in neurons and glial cells [6]. Insulin signaling begins when insulin binds to and activates the IR, and this results in the activation and phosphorylation of other substrates. The IR’s tyrosine kinase activity phosphorylates insulin receptor substrate-1 (IRS-1), and phosphorylated IRS-1 has binding sites for various downstream signaling partners. Of these, phosphatidylinositide-3-kinase (PI3K) is important. Activated PI3K activates phosphoinositide-dependent protein kinase 1 (PDK1), which then activates AKT. A major target of PDK1 is glycogen synthase kinase-3 (GSK-3). When AKT phosphorylates GSK-3β or GSK-3α, GSK-3 is suppressed, and this results in glycogen synthesis [27].

Ultimately, obesity leads to a down-regulation of the downstream signaling pathways of insulin, and the down-regulation of insulin signaling reduces glucose metabolism. Additionally, previous studies have reported that obesity from a HFD decreases the activity of the hippocampal insulin signaling pathway, including the activity of IRS-1, PI3K, p-PDK1, p-AKT, and p-GSK-3β [28,29]. Similarly, in our study, the activity of the hippocampal insulin-signaling pathway, including the IR and downstream molecules, was reduced in the HFD group. The down-regulation of insulin signaling in the hippocampus may affect neuroplasticity because a sufficient supply of glucose cannot be established. According to previous studies, insulin regulates synaptic plasticity in the hippocampus [30], and HFD-induced obesity reduces hippocampal BDNF levels and neurogenesis, and this leads to impairments in cognitive function [31]. Deficiencies in hippocampal insulin signaling impair memory by impairing synaptic plasticity [32]. In our study, we found that there was a reduction in neurogenesis and in the levels of BDNF, trkB, and DCX. We also discovered that there was a deficit in hippocampal insulin signaling.

Conversely, exercise is one of the most effective methods for managing obesity, and it has a positive effect on brain function. It has been suggested that exercise alters the efficient and flexible regulation of neural circuits that support cognitive function in overweight children [33] and improves the decline in cognitive function that is caused by various brain diseases and brain damage [34,35]. In our study, the HFD and EX group showed reduced body weight, improved intracellular glucose absorbance, and improved cognitive function as compared with the HFD group. Here, we demonstrated that exercise affects the CNS as well as peripheral nerves and improves cognitive functions, including learning and memory, by alleviating hippocampal insulin resistance that is caused by obesity. In previous studies, exercise has been shown to increase insulin sensitivity in brains when insulin resistance is caused by streptozotocin (STZ) administration [36]. Furthermore, studies have reported that exercise improves spatial learning by enhancing hippocampal insulin signaling pathways (IR/pIR/pAKT) and glucose usage rates [37]. Accordingly, in our study, obesity caused impairments in hippocampal pIR/pPDK/pAKT and pGSK3β, which resulted in insulin resistance, but these impairments were improved by exercise. In particular, PI3K/AKT, which is produced by IR activation, is an important signaling pathway for synaptic plasticity [38], and GSK3β activity is essential for determining the direction of synaptic plasticity [39]. Previous studies have reported that exercise augments the phosphorylation of PI3K/AKT and GKS3α/β as well as the activity of insulin signaling, which negatively regulates GSK3β in the hippocampus [40,41]. In our study, exercise in the obese mice activated insulin-signaling pathways, especially those affecting the ratio of IR to p-IR, PI3K to p-PI3K, AKT to p-AKT, and GSK3β to p-GSK3β in the hippocampus. Brain insulin signaling is required for neuronal survival and maintenance of crucial brain functions, such as cognitive functions [42], and can both prevent and reverse the defects in the BDNF transport [43]. In addition, hippocampal neurogenesis is regulated by BDNF, and an increased BDNF level due to activated insulin signaling by exercise. Similarly, studies have reported that exercise improves obesity-induced reduction in the hippocampal BDNF level [44]. Exercise has been proposed to increase hippocampal cell proliferation and differentiation in obese and diabetic animal models [45]. Activation of hippocampal insulin signaling by exercise might increase hippocampal neuroplasticity including BDNF and neurogenesis in the presence of obesity.

## 5. Conclusions

Obesity results in insulin resistance in peripheral tissues and the CNS, particularly in the brain. Importantly, hippocampal insulin resistance causes a reduction in neuroplasticity and leads to impairment in cognitive functions. Conversely, improving insulin resistance by exercise, and thereby hippocampal glucose metabolism might improve insulin signaling and neuroplasticity resulting in the alleviation of cognitive dysfunction in obesity. From the results of the present study, we have concluded that exercise acts as a potential non-pharmacological aid that protects against obesity-induced decline in cognitive functions by improving hippocampal insulin signaling and neuroplasticity.

## Figures and Tables

**Figure 1 nutrients-11-01603-f001:**
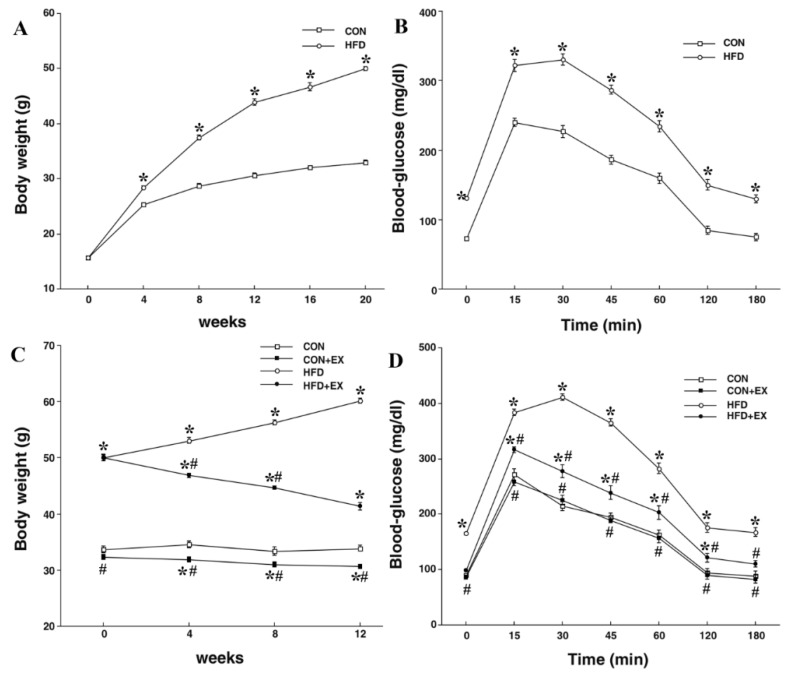
Effects of a high-fat diet (HFD) and treadmill exercise on body weight and blood glucose levels. Body weight (**A**) and blood glucose levels (**B**) after 20 weeks of a HFD. Body weight (**C**) and blood glucose levels (**D**) after 12 weeks of treadmill exercise. * Represents *p* < 0.05 as compared with the control (CON) group. # Represents *p* < 0.05 as compared with the high fat diet (HFD) group.

**Figure 2 nutrients-11-01603-f002:**
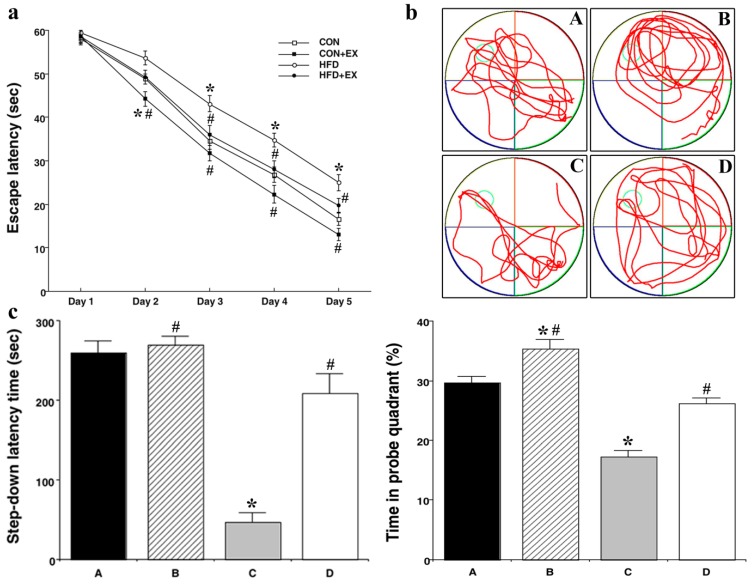
Effect of treadmill exercise on learning and memory using the Morris water maze and long-term memory using the step-down. Spatial learning (**a**) and working memory tracking (**b**) in the Morris water maze and long-term memory (**c**) in the step-down avoidance test. Control group **A**, control and exercise group **B**, high-fat diet group **C**, and high-fat diet and exercise group **D**. Data are expressed as the mean ± S.E.M. * Represents *p <* 0.05 as compared with the control (CON) group. # Represents *p <* 0.05 as compared with the high-fat diet (HFD) group.

**Figure 3 nutrients-11-01603-f003:**
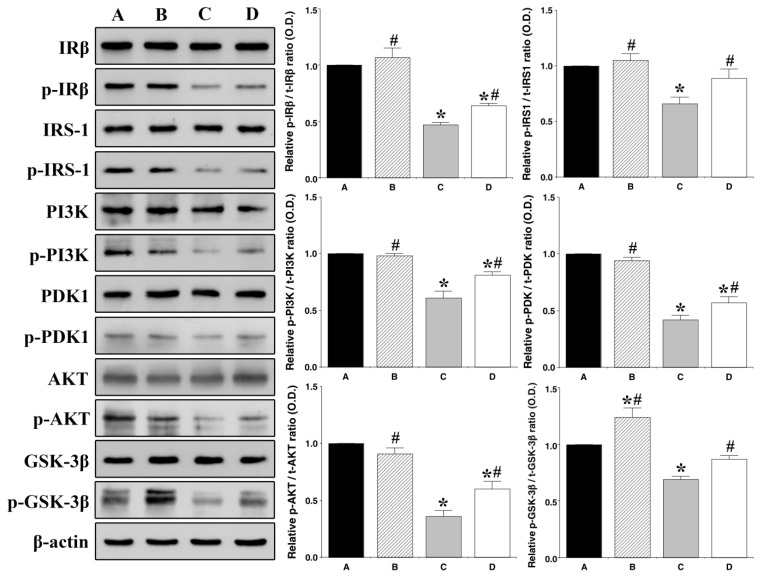
Effect of treadmill exercise on hippocampal insulin signaling. Control group **A**, control and exercise group **B**, high-fat diet group **C**, and high-fat diet and exercise group **D**. Data are expressed as the mean ± S.E.M. * Represents *p <* 0.05 as compared with the control (CON) group. # Represents *p <* 0.05 as compared with the high-fat diet (HFD) group.

**Figure 4 nutrients-11-01603-f004:**
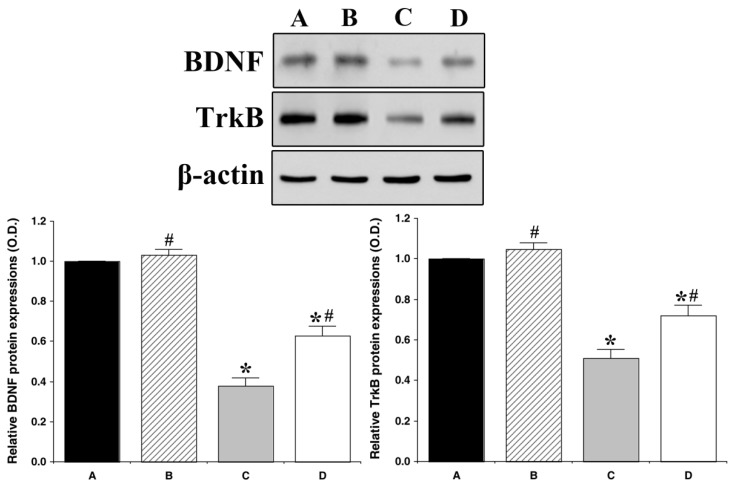
Effect of treadmill exercise on hippocampal BDNF and TrkB. Control group **A**, control and exercise group **B**, high-fat diet group **C**, and high-fat diet and exercise group **D**. Data are expressed as the mean ± S.E.M. * Represents *p <* 0.05 as compared with the control (CON) group. # Represents *p <* 0.05 as compared with the high fat diet (HFD) group.

**Figure 5 nutrients-11-01603-f005:**
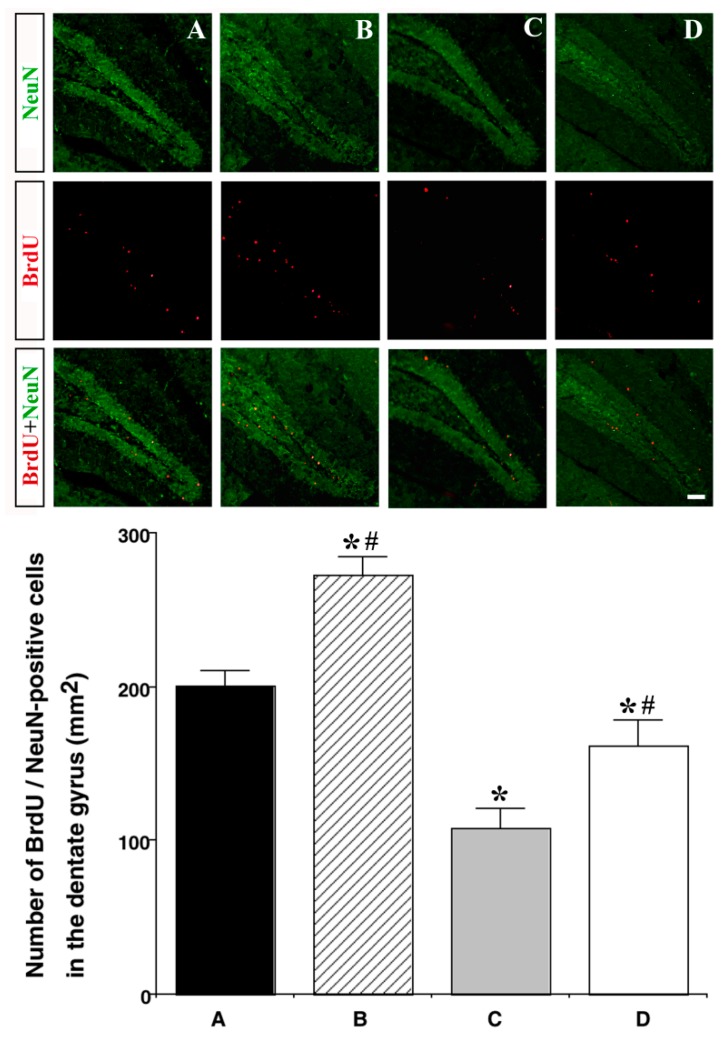
Effect of treadmill exercise on neurogenesis in the hippocampal dentate gyrus. (Above) Photomicrographs of BrdU/NeuN-double-positive cells. The scale bar represents 100 μm. (Below) The number of BrdU/NeuN-double-positive cells in each group. Control group **A**, control and exercise group **B**, high-fat diet group **C**, and high-fat diet and exercise group **D**. Data are expressed as the mean ± S.E.M. * Represents *p <* 0.05 as compared with the control (CON) group. # Represents *p <* 0.05 as compared with the high-fat diet (HFD) group.

**Figure 6 nutrients-11-01603-f006:**
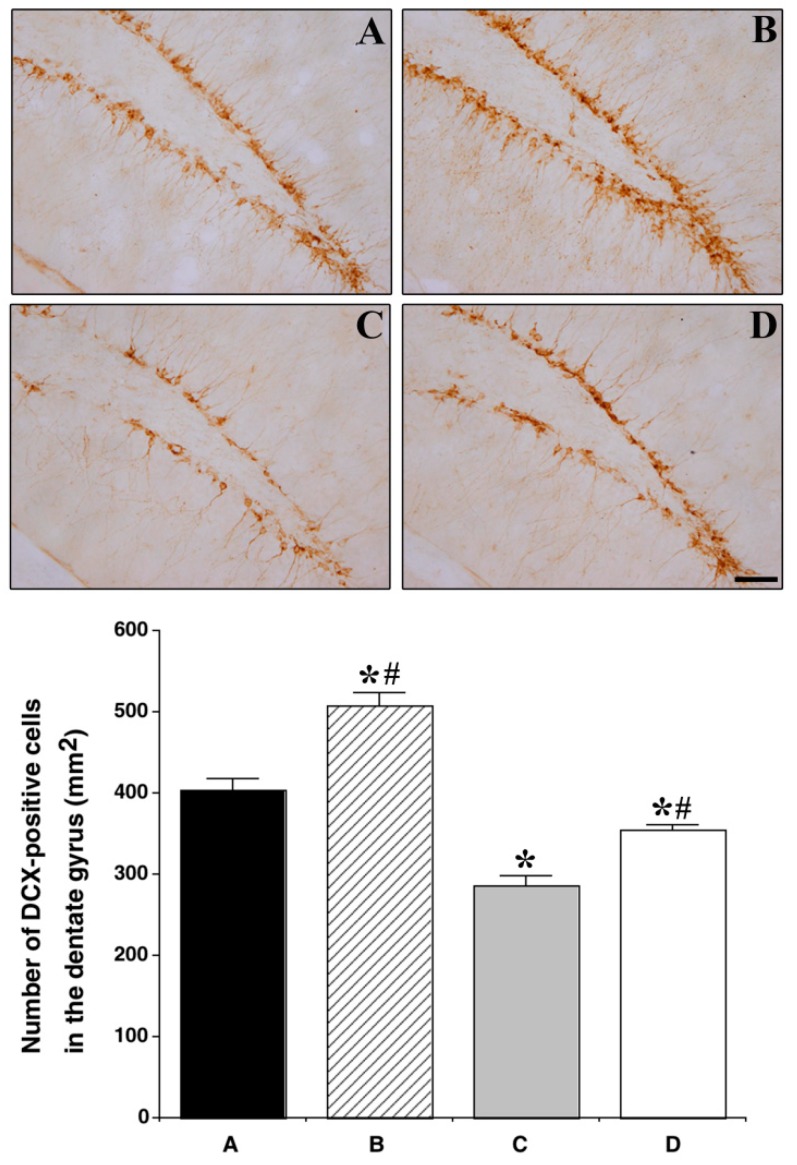
Effect of treadmill exercise on neurogenesis in the hippocampal dentate gyrus. (Above) Photomicrographs of DCX-positive cells. The scale bar represents 50 µm. (Below) The number of DCX-positive cells in each group. Control group **A**, control and exercise group **B**, high-fat diet group **C**, and high-fat diet and exercise group **D**. Data are expressed as the mean ± S.E.M. * Represents *p <* 0.05 as compared with the control (CON) group. # Represents *p <* 0.05 as compared with the high fat diet (HFD) group.

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
