# Peer review of "Exercise Alleviates Cognitive Functions by Enhancing Hippocampal Insulin Signaling and Neuroplasticity in High-Fat Diet-Induced Obesity"

_nutrients, 2019, doi:10.3390/nu11071603_

Round 1
Reviewer 1 Report
General Comment
Numerous studies identified that the obesity and/or Type 2 diabetes are risk factors of dementia including Alzheimer’s disease. On the other hand, habitual exercise prevents there non-communicable disease, and induces neurogenesis in brain. This study compared the effects of exercise training on cognitive function and insulin signaling among in exercise and high fat diet (HFD) mice. As the result, they demonstrated that the exercise training improves cognitive function and neurogenesis in HFD mice. There are very interesting and clear results. In addition, authors well discussed about the results. The reviewer curious to know follow points.
Minor comment
Authors shown the exercise induced neurogenesis in hippocampal dentate gyrus (Figure 5). I wonder, how about the CA1, and CA2. There regions also important for memory and cognitive function. If you have the results, please describe in results and/or discussion. Otherwise, please describe the importance of increase dentate gyrus cells (eg. function of dentate gyrus).
Authors hypothesized the insulin signaling is important for cognitive function. The hyperinsulinemia were found in HFD groups (HFD, and HFD + Ex, after 20 weeks). If you measured insulin concentration (baseline or during glucose tolerance test), please add the data.
Author Response
Review 1
Minor comment
1. Comments: Authors shown the exercise induced neurogenesis in hippocampal dentate gyrus (Figure 5). I wonder, how about the CA1, and CA2. There regions also important for memory and cognitive function. If you have the results, please describe in results and/or discussion. Otherwise, please describe the importance of increase dentate gyrus cells (eg. function of dentate gyrus).
Answers: We thank the reviewer for these suggestions. I know that neurogenesis in the adult brain of mammals occurs in two locations: 1) the subgranular zone of the lateral ventricles and 2) the subgranular zone of the dentate gyrus of the hippocampus. However, like the reviewer’s suggestion, neurogenesis occurs in the dentate gyrus and regions CA1, CA2, and CA3 of the hippocampus during development (2 or 3 weeks in rats or mice) However, our experimental animals were 36 weeks old.
Zhao et al (2008) Mechanisms and functional implications of adult neurogenesis. Cell, 132, 645-660
Kim et al (2006) Influence of prenatal noise and music on the spatial memory and neurogenesis in the hippocampus of developing rats. Brain and development, 28, 109-114.
According to the reviewer comments, we have included the role of the dentate gyrus in neurogenesis in the introduction (page 2, lines 94-96).
2. Comments: Authors hypothesized the insulin signaling is important for cognitive function. The hyperinsulinemia were found in HFD groups (HFD, and HFD + Ex, after 20 weeks). If you measured insulin concentration (baseline or during glucose tolerance test), please add the data
Answers: We appreciate your insightful comments. We did not measure insulin concentration, but we believe that hyperinsulinemia can be indirectly measured through the use of a glucose tolerance test. In future studies, we will incorporate the reviewer’s advice.
Muller et al (2011) Exercise increases insulin signaling in the hippocampus: physiological effects and pharmacological impact of intracerebroventricular insulin administration in mice. Hippocampus 21: 1082-1092.
We advise that manuscript has been checked by a native English-speaking copy editor who is familiar with the concepts addressed in the manuscript.
Reviewer 2 Report
I have no further requests.
Author Response
We advise that manuscript has been checked by a native English-speaking copy editor who is familiar with the concepts addressed in the manuscript.
Reviewer 3 Report
Park et al. analyzed the effects of exercise on the cognitive impairment induced by the high-fat diet, and its relationship between hippocampal insulin signaling. This manuscript was well-written. However, the biggest problem of this study was its novelty, because almost the same results were previously published.
Major points
1) The similar results were previously published by different journals.
Fig. 2 was similar to Park HS et al., Metab Brain Dis, 2018.
Fig. 4, 5, 6 were similar to Kim TW et al., J Exerc Rehabil, 2016.
Furthermore, improvement of insulin-Akt signaling by exercise in the hippocampus was reported by Muller AP et al., Neurochem Res, 2008 and Jeong JH et al., J Exerc Nutrition Biochem, 2018.
Fig. 1 was the general effects of a high-fat diet.
In general, the beneficial effects of exercise on cognitive impairment and on insulin resistance were well-recognized. Thus, the overall novelty of this study was quite low.
Minor points
2) The abbreviation for CNS and IR were described twice in line 40, 56 and in line 37, 39, respectively.
3) The cat. number of used antibodies were required.
4) In line 135, which phosphorylation site of Akt was analyzed?
5) In Fig. 1, the explanation of # and * was required.
Author Response
Review 3
Major points
Comments: 1) The similar results were previously published by different journals.
Fig. 2 was similar to Park HS et al., Metab Brain Dis, 2018.
Fig. 4, 5, 6 were similar to Kim TW et al., J Exerc Rehabil, 2016.
Furthermore, improvement of insulin-Akt signaling by exercise in the hippocampus was reported by Muller AP et al., Neurochem Res, 2008 and Jeong JH et al., J Exerc Nutrition Biochem, 2018.
Fig. 1 was the general effects of a high-fat diet.
In general, the beneficial effects of exercise on cognitive impairment and on insulin resistance were well-recognized. Thus, the overall novelty of this study was quite low.
1. Answers: First, thank you for knowing our error part by your detailed review. We apologize for the original sentence was erroneously written. We have revised short-term memory to long-term memory.
Metab Brain Dis and J Exerc Rehabil is one of our previous papers. As the reviewer commented, the studies are similar; however, the purpose of this study was different.
The article in Metab Brain Dis focuses on mitochondrial functions and apoptosis in the hippocampus as a result of insulin resistance (Insulin tolerance test) that was due to obesity, and the article in J Exerc Rehabil focuses on hippocampal cell differentiation.
The main purpose of a high-fat diet was to establish an obesity animal model.
As the reviewer commented, I believe that the novelty of this study may be low. However, this study is expected to further support scientific evidence of the direct effects of exercise on hippocampal insulin signaling, neurogenesis, neurotrophic factors, and cognitive functioning in patients with obesity.
Minor points
2. Comment: The abbreviation for CNS and IR were described twice in line 40, 56 and in line 37, 39, respectively.
Answers: Thank you for your comment in detailed. I have removed it
3. Comment: The cat. number of used antibodies were required.
Answers: According to the review comment, I have added number of antibody
4. Comment: In line 135, which phosphorylation site of Akt was analyzed?
Answers: According to the reviewer’s comments, I have added the phosphorylation site of Akt.
5. Comment: In Fig. 1, the explanation of # and * was required
Answers: According to the reviewer’s comments, we have added explanations of # and * in Fig. 1.
We advise that manuscript has been checked by a native English-speaking copy editor who is familiar with the concepts addressed in the manuscript.
Round 2
Reviewer 3 Report
As the authors noticed, almost the same results were previously published by the authors. I understand that the purpose of this study was different from previous studies. However, the novelty of the paper should be evaluated by the results. The authors should add the novel aspects of exercise on cognitive impairment and on insulin resistance.
Author Response
Comment: As the authors noticed, almost the same results were previously published by the authors. I understand that the purpose of this study was different from previous studies. However, the novelty of the paper should be evaluated by the results. The authors should add the novel aspects of exercise on cognitive impairment and on insulin resistance.
Answers: We thank the reviewer for these suggestions again. According to the reviewer’s comments, we have modified the Introduction and Discussion sections as follows:
Introduction
Especially, dysfunctions in the hippocampal insulin signaling might be responsible for cognitive impairment [15]. The significant risk factors for the preliminary-diabetic stage of insulin resistance and type-II diabetes, such as obesity and physical inactivity are associated with alterations in the brain and increase the risk of developing dementia [16,17,18].
15. Biessels,G.J.; Reagan, L.P. Hippocampal insulin resistance and cognitive dysfunction. Nat Rev Neurosci 2015, 16, 660-671.
16. Luchsinger, J.A. Adiposity, hyperinsulinemia, diabetes and Alzheimer's disease: an epidemiological perspective. Eur J Pharmacol 2008, 5, 9-29.
17. Norton,S.; Matthews, F.E.; Barnes, D.E.; Yaffe, K.; Brayne, C. Potential for primary prevention of Alzheimer's disease: an analysis of population-based data. Lancet Neurol 2014, 13, 788-794.
18. Kiliaan, A.J.; Arnoldussen, I.A.; Gustafson, D.R. Adipokines: a link between obesity and dementia? Lancet Neurol 2014, 13, 913-923.
Discussion
Previous studies have reported that exercise augments the phosphorylation of PI3K/AKT and GKS3α/ β [40], as well as the activity of insulin signaling, which negatively regulates GSK3 β in hippocampus [41]. In our study, exercise in the obese mice activated insulin-signaling pathways, especially those affecting the ratio of IR to p-IR, PI3K to p-PI3K, AKT to p-AKT, and GSK3β to p-GSK3β in the hippocampus. Brain insulin signaling is required for neuronal survival and maintenance of the crucial brain functions, such as cognitive functions [42], and can both prevent and reverse the defects in the BDNF transport [43]. In addition, hippocampal neurogenesis is regulated by BDNF, and an increase BDNF level due to activated insulin signaling by exercise [44]. Similarly, studies have reported that exercise improves obesity-induced reduction in the hippocampal BDNF level [45]. Exercise has been proposed to increase hippocampal cell proliferation and differentiation in obese and diabetic animal models [46]. Activation of hippocampal insulin signaling by exercise might increase hippocampal neuroplasticity including BDNF and neurogenesis in the presence of obesity.
40. Um, H.S.; Kang, E.B.; Koo, J.H.; Kim, H.T.; Jin-Lee.; Kim, E.J.; Yang, C.H.; An, G.Y.’ Cho, I.H.; Cho, J.Y. Treadmill exercise represses neuronal cell death in an aged transgenic mouse model of Alzheimer's disease. Neurosci Res 2011, 69, 161-73.
41. Zang, J.; Liu, Y.; Li, W.; Xiao, D.; Zhang, Y.; Luo, Y.; Liang, W.; Liu, F.; Wei, W. Voluntary exercise increases adult hippocampal neurogenesis by increasing GSK-3β activity in mice. Neuroscience 2017, 354, 122-135.
42. Kleinridders, A.; Ferris, H.A.; Cai, W.; Kahn, C.R.. Insulin action in brain regulates systemic metabolism and brain function. Diabetes 2014, 63, 2232-2243.
43. Takach, O.; Gill, T.B.; Silverman, M.A. Modulation of insulin signaling rescues BDNF transport defects independent of tau in amyloid-β oligomer-treated hippocampal neurons. Neurobiol Aging 2015, 36, 1378-1382.
44. Lee, K.; Martin, B.; Maudsley, S.; Golden, E.; Cutler, R.G.; Mattson, M.P. Voluntary exercise and caloric restriction enhance hippocampal dendritic spine density and BDNF levels in diabetic mice. Hippocampus 2009, 19, 951-961.
5. Conclusions
Obesity results in insulin resistance in the peripheral tissues and the CNS, particularly in the brain. Importantly, hippocampal insulin resistance causes a reduction in neuroplasticity and leads to impairment in cognitive functions. Conversely, improving insulin resistance by exercise, and thereby hippocampal glucose metabolism might improve insulin signaling and neuroplasticity result in alleviates cognitive dysfunction in obesity. From the results of the present study, we conclude that exercise acts as a potential non-pharmacological aid that protects against obesity-induced decline in cognitive functions by improving hippocampal insulin signaling and neuroplasticity.